# *N*-Containing α-Mangostin Analogs via Smiles Rearrangement as the Promising Cytotoxic, Antitrypanosomal, and SARS-CoV-2 Main Protease Inhibitory Agents

**DOI:** 10.3390/molecules28031104

**Published:** 2023-01-22

**Authors:** Nan Yadanar Lin Pyae, Arnatchai Maiuthed, Wongsakorn Phongsopitanun, Bongkot Ouengwanarat, Warongrit Sukma, Nitipol Srimongkolpithak, Jutharat Pengon, Roonglawan Rattanajak, Sumalee Kamchonwongpaisan, Zin Zin Ei, Preedakorn Chunhacha, Patcharin Wilasluck, Peerapon Deetanya, Kittikhun Wangkanont, Kowit Hengphasatporn, Yasuteru Shigeta, Thanyada Rungrotmongkol, Supakarn Chamni

**Affiliations:** 1Pharmaceutical Sciences and Technology Program, Faculty of Pharmaceutical Sciences, Chulalongkorn University, Bangkok 10330, Thailand; 2Department of Pharmacognosy and Pharmaceutical Botany, Faculty of Pharmaceutical Sciences, Chulalongkorn University, Bangkok 10330, Thailand; 3Natural Products and Nanoparticles Research Unit (NP2), Chulalongkorn University, Bangkok 10330, Thailand; 4Department of Pharmacology, Faculty of Pharmacy, Mahidol University, Bangkok 10400, Thailand; 5Centre of Biopharmaceutical Science for Healthy Ageing, Faculty of Pharmacy, Mahidol University, Bangkok 10400, Thailand; 6Department of Biochemistry and Microbiology, Faculty of Pharmaceutical Sciences, Chulalongkorn University, Bangkok 10330, Thailand; 7National Center for Genetic Engineering and Biotechnology (BIOTEC), National Science and Technology Development Agency, Pathum Thani 12120, Thailand; 8Center of Excellence for Molecular Biology and Genomics of Shrimp, Department of Biochemistry, Faculty of Science, Chulalongkorn University, Bangkok 10330, Thailand; 9Center of Excellence for Molecular Crop, Department of Biochemistry, Faculty of Science, Chulalongkorn University, Bangkok 10330, Thailand; 10Center for Computational Sciences, University of Tsukuba, 1-1-1 Tennodai, Tsukuba 305-8577, Ibaraki, Japan; 11Center of Excellence in Biocatalyst and Sustainable Biotechnology, Department of Biochemistry, Faculty of Science, Chulalongkorn University, Bangkok 10330, Thailand; 12Program in Bioinformatics and Computational Biology, Graduate School, Chulalongkorn University, Bangkok 10330, Thailand

**Keywords:** α-mangostin analogs, smiles rearrangement, intramolecular nucleophilic aromatic substitution reaction, isosteric replacement, anticancer, antimalarial, antitrypanosomal, SARS-CoV-2 main protease inhibitor, fragment molecular orbital (FMO) method

## Abstract

New *N*-containing xanthone analogs of α-mangostin were synthesized via one-pot Smiles rearrangement. Using cesium carbonate in the presence of 2-chloroacetamide and catalytic potassium iodide, α-mangostin (**1**) was subsequently transformed in three steps to provide ether **2**, amide **3**, and amine **4** in good yields at an optimum ratio of 1:3:3, respectively. The evaluation of the biological activities of α-mangostin and analogs **2**–**4** was described. Amine **4** showed promising cytotoxicity against the non-small-cell lung cancer H460 cell line fourfold more potent than that of cisplatin. Both compounds **3** and **4** possessed antitrypanosomal properties against *Trypanosoma brucei rhodesiense* at a potency threefold stronger than that of α-mangostin. Furthermore, ether **2** gave potent SARS-CoV-2 main protease inhibition by suppressing 3-chymotrypsinlike protease (3CL^pro^) activity approximately threefold better than that of **1**. Fragment molecular orbital method (FMO–RIMP2/PCM) indicated the improved binding interaction of **2** in the 3CL^pro^ active site regarding an additional ether moiety. Thus, the series of *N*-containing α-mangostin analogs prospectively enhance druglike properties based on isosteric replacement and would be further studied as potential biotically active chemical entries, particularly for anti-lung-cancer, antitrypanosomal, and anti-SARS-CoV-2 main protease applications.

## 1. Introduction

α-Mangostin is a major natural oxygenated and prenylated xanthone isolated from the pericarp of mangosteen (*Garcinia mangostana*) [1]. Various in vitro and in vivo studies have reported that α-mangostin exhibits a wide range of medicinal properties as a potential lead for anticancer, antioxidant, antiviral, antifungal, antiobesity, neuroprotection, insecticidal, anti-inflammatory, antibacterial, and antiprotozoal applications [1,2]. α-Mangostin displays interesting anticancer and cytotoxic properties toward several cancer cell lines, including leukemia, lung, breast, ovary, liver, skin, brain, prostate, colon, and cervical cancers [3]. Moreover, the antimicrobial activity of α-mangostin had been studied for both Gram-positive and Gram-negative bacteria, including the human-life-threatening sepsis of *Staphylococcus aureus* [4,5,6], skin and nosocomial infections of *Staphylococcus epidermidis* [7,8], and urinary tract infections of *Escherichia coli* [9,10]. The antimalarial activities of α-mangostin against *Plasmodium falciparum* [11] and antitrypanosomal activities against *Trypanosoma brucei* [12,13], which cause endemic parasitic infections such as malaria and human African trypanosomiasis (also known as sleeping sickness), respectively, had also been investigated. Although α-mangostin has potential therapeutic effects, there is currently no clinically approved information because of its extreme hydrophobicity, which limits its solubility and stability in aqueous systems, leading to low oral bioavailability [14,15,16].

Various series of α-mangostin derivatives had been synthesized by modifying phenolic hydroxyl groups via alkylation and acetylation [5,17,18,19] to improve α-mangostin’s druglike properties (e.g., solubility, permeability, metabolic stability, and transporter effects) and endow it with better biological properties. Interestingly, bioisosteric replacement by exchanging atoms such as carbon and oxygen with a nitrogen atom has been reported as an efficient strategy for enhancing α-mangostin’s physicochemical properties and intramolecular and intermolecular interactions and the pharmacological profiles of lead compounds [20] such as the discovery of *N*-containing analogs of vancomycin and vinblastine [21].

Series of *N*-containing α-mangostin analogs were previously prepared on the basis of the Mannich reaction [22,23] and carbamate formation [24,25] (Figure 1). Importantly, the bioisosteric replacement of an oxygen atom of α-mangostin at C–3 and C–6 positions, which are phenolic moieties, was unexplored. In this regard, Smiles rearrangement is an intramolecular nucleophilic aromatic substitution reaction. The rearrangement occurs when a nucleophile such as alcohol, phenol, amine, amide, and sulfonamide displaces an aromatic electrophile such as ether, sulfide, sulfoxide, or sulfone under basic conditions [26,27]. This reaction is a convenient one-pot transformation of phenols to anilines involving nucleophilic substitution, rearrangement, and hydrolysis [28]. Recently, the transformation of chrysin, a natural flavonoid, to 7-aminochrysin derivatives by Smiles rearrangement was reported [29]. Therefore, this study aims to investigate the novel series of semisynthetic α-mangostin analogs via Smiles rearrangement. Furthermore, the cytotoxic, antibacterial, antimalarial, antitrypanosomal, and anti-SARS-CoV-2 main protease activities of new *N*-containing α-mangostin analogs were evaluated.

## 2. Results and Discussion

### 2.1. One-Pot Synthesis of N-Containing α-Mangostin Analogs via Smiles Rearrangement

α-Mangostin was isolated from dried mangosteen pericarp powder as the major natural xanthone. From the isolation and purification processes, 15.8 g of α-mangostin was obtained as a yellow powder, and the isolation yield was 3.2% based on the weight of the dried plant material. The structural identification of α-mangostin was confirmed using ^1^H and ^13^C nuclear magnetic resonance (NMR) spectroscopy [30,31]. The isolated α-mangostin was employed as the starting material for the investigation of the one-pot synthesis of *N*-containing α-mangostin analogs via Smiles rearrangement. The transformation of the hydroxyl groups of the phenol moiety of α-mangostin to primary amines via Smiles rearrangement requires the use of 2-chloroacetamide as an activating agent, potassium iodide (KI) as a catalyst, and *N*,*N*-dimethyl formamide (DMF) as a high boiling point solvent in the presence of a suitable alkaline base and heat to facilitate tandem nucleophilic substitution, rearrangement, and hydrolysis reaction (Figure 1). The Smiles rearrangement of α-mangostin (**1**) commenced using 2.5 equivalent of potassium carbonate (K_2_CO_3_) as a base and 0.2 equivalent of KI as a catalyst to afford the series of *N*-containing α-mangostin analogs, including ether **2**, amide **3**, and amine **4**, which were sequentially obtained through nucleophilic substitution, rearrangement, and hydrolysis, respectively. Interestingly, the regioselectivity of Smiles rearrangement at the hydroxyl group at the C–3 and C–6 positions was observed. The proposed mechanism for the Smiles rearrangement of α-mangostin (**1**) yielding the series of *N*-containing α-mangostin analogs **2**−**4** was illustrated in Figure 2. The transformation of hydroxyl group at C–6 was observed as a major product at 90 °C. Both hydroxyl groups at C–3 and C–6 positions were modified at higher temperature (150 °C). However, the hydroxyl group at C–1 was intact because of the steric hindrance from an isoprene substituent at the C–2 and C–8 positions and stabilization via hydrogen bonding between a hydroxyl group at C–1 and ketone at C–9.

Compounds **2**–**4** were structurally characterized using spectroscopic techniques (Figure 2, and see Appendix A for the NMR spectra). Besides the characteristic structures of xanthone **1**, including a hydroxyl group at C–1, a prenyl group, an aromatic proton at C–4 and C–5, and a methoxy group at C–7, compounds **2**–**4** displayed ^1^H and ^13^C chemical shifts of additional motifs at C–3 and C–6. Compound **2**, derived from the nucleophilic substitution step, showed the characteristic peaks of acetamide substituent from methylene proton (2′–CH_2_) as the singlet at 4.23 ppm for ^1^H and from primary amide carbonyl at 171.8 ppm and methylene carbon at 63.1 ppm for ^13^C chemical shifts. A significant upfield chemical shift of C–6 at 144.4 ppm was observed. Meanwhile, the characteristic proton chemical shifts of compound **3**, a product of the rearrangement step, included the broad singlet of amide hydrogen (1′–NH) at 6.85 ppm (overlapped) and the singlet of methylene proton (3′–CH_2_) at 4.65 ppm. Regarding the ^13^C-NMR spectrum, amide **3** displayed secondary amide carbonyl at 170.0 ppm and methylene carbon at 68.3 ppm. An upfield chemical shift at an aromatic C–6 of amide **3** was observed. Compound **4**, the final product of the Smiles rearrangement obtained from the hydrolysis step, showed the characteristics of proton chemical shifts of amine (NH_2_) at C–3 and C–6 at 5.37 and 5.69 ppm, respectively, along with chemical shifts of C–3 and C–6 aromatic carbons at 154.3 and 148.7 ppm, respectively. Upfield chemical shifts of aromatic carbon were clearly observed at C–2/C–3 and C–5/C–6.

### 2.2. Optimization of the Smiles Rearrangement of α-Mangostin (***1***)

The reaction conditions (Table 1) were optimized to improve the synthesis of amine **4**. The screening of the suitable alkaline base and sufficient amount of 2-chloroacetamide and KI along with proper heating approach and time led to the efficient Smiles rearrangement of **1**. Upon optimization, the reaction concentration was controlled at 0.05 mM, and a stoichiometric amount of alkaline base was used at 2.5 equivalent. The initial trial using K_2_CO_3_ gave a mixture of compounds **2**:**3**:**4** with 46% yield at a ratio of 2:2:1 (Table 1, entry 1). This result suggests that **1** was converted to **2** and **3** through nucleophilic substitution and rearrangement at moderate conversion and slightly went to hydrolysis.

According to the low transformation of **1** toward **4**, the increasing amounts of 2-chloroacetamide and KI were investigated. The addition of an equal amount of 2-chloroacetamide to K_2_CO_3_ at 2.5 equivalent could facilitate the later steps of the Smiles rearrangement involving rearrangement and hydrolysis to provide **3** and **4** at higher ratios with 53% yield (Table 1, entry 2). Using 2.5 equivalent of 2-chloroacetamide together with an increased amount of KI to 0.5–1 equivalent afforded an improved yield (54–65%). However, nucleophilic substitution and rearrangement were more favorable than hydrolysis, even though the reaction time was expanded to 10 h (Table 1, entries 3–5). The use of a microwave apparatus with 2.5 equivalent of 2-chloroacetamide and a catalytic amount of KI significantly produced **4** at a good ratio (Table 1, entry 6). However, the overall yield was unsatisfactory.

The use of cesium carbonate (Cs_2_CO_3_) as a base in the presence of 2-chloroacetamide (1.2 equivalent) and catalytic KI (20%) gave an excellent overall yield at 85%, and the rearrangement and hydrolysis steps were favorable. (Table 1, entry 7). Interestingly, an increased amount of 2-chloroacetamide (2.5 equivalent) gave a moderate yield (50%), and amine **4** became a major product (entry 8). The use of a stochiometric amount of KI specifically facilitated the nucleophilic substitution and rearrangement steps (entry 9). Under microwave, amine **4** was obtained as a major product with a low yield. The Smiles rearrangement of **1** with potassium hydroxide (KOH), a strong alkaline base, produced low overall yields compared with those of the K_2_CO_3_ and CsCO_3_ conditions. The reaction containing KOH, 2-chloroacetamide (1.2 equivalent), and catalytic KI (20%) provided amine **4** at a significantly high ratio (entry 11). However, the increased amount of 2-chloroacetamide (2.5 equivalent) mediated the rearrangement step to obtain amide **3** as the major product (entry 12). The condition with a stochiometric amount of KI led to an unsatisfactory ratio of compounds **2**:**3**:**4** (entry 13). Heating through microwave delivered amine **4** as a major product with low yield (entry 14).

The screening of the facile reaction conditions for the Smiles rearrangement of α-mangostin (**1**) was described. The equal equivalent between the alkaline base and 2-chloroacetamide in the presence of a catalytic amount of KI facilitated the three-step one-pot Smiles rearrangement and gave a moderate overall yield. Moreover, the stochiometric amount of KI preferably drove the first two steps involving nucleophilic substitution and rearrangement. The results also suggest that the use of a strong alkaline base such as KOH and microwave conditions gave a low yield. Interestingly, the use of equal amounts of KOH and 2-chloroacetamide along with catalytic KI (20%) delivered compound **3** as the major product. Therefore, the optimum condition comprised the equal equivalent of Cs_2_CO_3_ and 2-chloroacetamide along with catalytic KI (20%) to obtain a satisfactory overall yield and a ratio of compound **4**.

Herein, we report for the first time the chemical and biological aspects of *N*-containing α-mangostin analogs **3** and **4**. As regards the isosteric replacement of the hydroxyl group at C–3 and C–6, compound **3** contained a 2-hydroxy acetamide substituent at C–3, whereas compound **4** embedded amine groups at C–3 and C–6 that would revise its druglike properties. The prediction of the absorption, distribution, metabolism, and excretion (ADME) parameters of the *N*-containing analogs **2**–**4** illustrated that compounds **2** and **3** showed improved flexibility, H-bond acceptors, and H-bond donors that exceed that of the parent **1**. In addition, compounds **3** and **4** had enhanced solubilities while maintaining their bioavailability score. The predicted pharmacokinetic properties also support the possible pharmaceutical application of *N*-containing α-mangostin analogs **3** and **4** (see Appendix A) [32,33,34].

### 2.3. Evaluation of Cytotoxicity

The cytotoxicities of compounds **1**–**4** were evaluated against several cancer cell lines, including non-small-cell lung cancer (H292 and H460), brain cancer (SW-1088 and U87-MG), breast cancer (MDA-MB-231 and MCF-7), hepatocellular carcinoma (HuH-7 and HepG2), and colon cancer (HT-29 and HCT-116) cell lines, along with kidney epithelial (Vero) cells as a normal cell line. The cytotoxicity values were compared with that of cisplatin as a positive control. The results in Table 2 show that α-mangostin (**1**) exhibited the most interesting cytotoxicity against all types of cancer cell lines with more advanced potency than that of cisplatin, a first-line chemotherapy drug. From the cytotoxicity data, α-mangostin inhibited the HT-29 colon cancer cell line at IC_50_ 12.28 ± 1.32 µM, which was stronger than that of cisplatin by at least eightfold. Compound **2**, an ether analog of **1** showed diminished cytotoxicity against all types of cancer cell lines. Compound **3**, an amide analog of **1**, contained a bioisosteric replacement by exchanging an oxygen atom for a nitrogen atom at C–6 and displayed noncytotoxicity (>200 µM) against non-small-cell lung cancer cell lines. However, amide **3** exhibited stronger cytotoxic activities against brain, breast, hepatocellular, and colon cancer cell lines than those of cisplatin but a slightly weaker potency than that of **1**, a mother compound. Interestingly, compound **4**, an amine analog of **1** with isosteric atoms by replacing oxygen atoms with nitrogen atoms at both C–1 and C–6, possessed selective cytotoxicity against the H460 non-small-cell lung cancer cell line at IC_50_ 13.89 ± 2.55 µM, which was more potent than those of α-mangostin and cisplatin by twofold and fourfold, respectively. Furthermore, compounds **1** and **3** showed cytotoxic activities against Vero cells at EC_50_ 22.00 ± 1.44 µM and 26.48 ± 0.59 µM respectively, whereas compounds **2** and **4** had low cytotoxicity (see Appendix A). Thus, amide **3** and diamine **4**, which are the bioisosteric analogs of **1**, would be further studied to disclose their druglike properties and anticancer mechanisms.

### 2.4. Evaluation of Antibacterial Activities

Among the compounds in this series, α-mangostin (**1**) showed the strongest antibacterial activity against the tested Gram-positive bacteria, *S. aureus*, *S. epidermidis*, *Kocuria rhizophila*, and *Bacillus subtilis*, with minimal inhibitory concentration (MIC) values of 1.22, 0.61, 0.61, and 1.22 µM, respectively. The MIC values of α-mangostin against *K. rhizophila* and *B. subtilis* were equal to those of vancomycin, whereas the activity against *S. epidermidis* was better than that of vancomycin. The series of *N*-containing α-mangostin analogs **2**–**4** did not show antibacterial activity against *S. aureus*. Meanwhile, the amine analog **4** showed activity against *S. epidermidis*, *K. rhizophila*, and *B. subtilis* with MIC values of 19.53, 4.88, and 39.06 µM, respectively, whereas analog **2** showed activity against *S. epidermidis*, *K. rhizophila*, and *B. subtilis*, with MIC values of 78.13, 9.76, and 312.5 µM, respectively. Compound **3** showed antibacterial activity against *K. rhizophila* with a MIC value of only 78.13 µM, and no activity was observed against *S. aureus*, *S. epidermidis*, and *B. subtilis*. All compounds in this series showed negative results against Gram-negative bacteria (*Klebsiella pneumoniae* and *E. coli*) at the final concentration of 2.5 mM (Table 3).

The antibacterial activity of **1** has been reported against *S. aureus*, methicillin-resistant *S. aureus*, and *B. cereus*, with MIC values ranging from 0.78 to 1.56 µg/mL [35]. The antibacterial mechanism of **1** involved interaction with bacterial membranes, leading to membrane disruption and bactericidal action. α-Mangostin (**1**) also showed activity against vancomycin-resistant Enterococci (with a MIC value of 6.25 µg/mL) and synergism with gentamicin [36]. Upon the comparison of antibacterial activities, the *N*-containing α-mangostin analogs **2**–**4** showed low activity compared with that of **1**. Overall, amine **4** showed interesting inhibition against *K. rhizophila*, where its inhibition value was stronger than those of acetamides **2** and **3** by up to twofold and 16-fold, respectively.

### 2.5. Evaluation of Antimalarial and Antitrypanosomal Activities

The biological activities of compounds **1**–**4** focusing on endemic parasitic infections were explored. For malaria, the antimalarial activity was evaluated against 3D7 *P. falciparum*, a wild-type drug-sensitive strain. For human African trypanosomiasis, antitrypanosomal activity was evaluated against *Trypanosoma brucei rhodesiense*. α-Mangostin (**1**) and its *N*-containing α-mangostin analogs (**2**–**4**) exhibited concentration-dependent inhibitory activities against both parasites (Table 4).

Compounds **1**–**3** showed moderate antimalarial activity at the inhibitory concentrations (IC_50_) of 5.45, 7.79, and 4.25 µM, respectively, whereas amine **4** possessed low activity at an IC_50_ of 13.00 µM. For the antitrypanosomal activity, α-mangostin (**1**) and ether **2** had moderate antitrypanosomal inhibitory potency at similar inhibitory values (IC_50_) of 6.74 and 7.04 µM, respectively, whereas amide **3** and amine **4** showed inhibitory concentrations (IC_50_) at 2.68 and 2.32 µM, respectively. Interestingly, amide **3** and amine **4** exhibited approximately threefold the potency of the mother compound **1**. These results suggest the replacement of the oxygenated motif; in this case, the hydroxy group was changed to a nitrogen-containing substituent, such as amide, and amine revised the biological activity with respect to the compound’s flexibility, H-bond acceptors, H-bond donors, and solubility, which can enhance cell membrane permeation and target protein interactions. Among the compounds in this series, amide **3** and amine **4** had the most interesting antitrypanosomal activity but mild inhibitory levels compared with that of pentamidine, the positive control. The new compounds **3** and **4** could be further studied for the development of antitrypanosomal agents, notably compound **4** showed selectivity toward *Trypanosoma* than *Plasmodium* parasites.

### 2.6. Evaluation of SARS-CoV-2 Main Protease (3CL^pro^) Inhibitory Activities

The preliminary screening of anti-SARS-CoV-2 targeted 3-chymotrypsinlike protease (3CL^pro^) of compounds **1**–**4**, evaluated according to the 8-anilinonaphthalene-1-sulfonate binding assay using E(EDANS)TSAVLQSGFRK(DABCYL) as the fluorogenic substrate compared with rutin as the positive control [37]. Relative protease activities treated with 100 μM of α-mangostin (**1**) and *N*-containing α-mangostin analogs **2**–**4** were obtained (Figure 3A). Compounds **1**–**4** were tested at a concentration of 0–500 μM (Figure 3B). The results of the relative 3CL^pro^ activities were calculated to IC_50_ values (Table 5). α-Mangostin (**1**) and 3,6 diamine-α-mangostin **4** exhibited similar inhibitory activity at 61.6 ± 1.1 and 63.8 ± 1.2 μM, respectively. Compound **2** showed potent inhibition at 24.6 ± 1.1 μM, which was threefold better than mother compound **1**. However, compound **3** lost the activity of 3CL^pro^ inhibition (IC_50_ at 325.1 ± 1.3 μM). 

### 2.7. Interaction Energy Profile of the Potent Compounds toward SARS-CoV-2 3CL^pro^

The binding mechanism and interaction of our potent compounds, α-mangostin (**1**) and *N*-containing α-mangostin analogs **2** and **4**, against SARS-CoV-2 3CL^pro^ were evaluated using the fragment molecular orbital (FMO) method. The pair interaction energy (PIE) and its decomposed analysis (PIEDA) at the FMO-RIMP2/PCM level of theory (Figure 4A) refer to the binding interaction of a ligand to its neighboring residues and electronic energy decomposed to each fragment. The binding pattern of the xanthone core structure of these compounds is likely aligned into the same binding pocket (S1, S2, and S1′) at the 3CL^pro^ active site (Figure 4B). The similar arrangement of these compounds in the active site is congruent to one of the best binding patterns of rubraxanthone in the previous report [39]. This core structure at the methoxy group at C–7 and the prenyl group at C–8 mainly interacted with the residues 41–45 in the S2 subpocket by dispersion ($E_{\mathrm{ij}}^{\mathrm{DI}}$) and charge-transfer and mix terms ($E_{\mathrm{ij}}^{\mathrm{CT}+\mathrm{mix}}$), referring to the hydrophobic interactions. Interestingly, the *N*-containing compound **2** showed interaction energy (PIEDA^total^ = −106.23 kcal/mol) significantly lower than its parent compound **1** (−72.09 kcal/mol) and analog **4** (−69.35 kcal/mol) due to the additional substituent containing an ended primary amide group at C–6 interacting to M49, L50, Y54, and R188 (Figure 4A) with three H-bond formations at M49, Y54 and D187, Figure 4B). For compound **4**, the substituted amine moieties interacted with H41, E166, and L167 mainly via electrostatic contribution, $E_{\mathrm{ij}}^{\mathrm{ES}}$. The catalytic residue H41, and E166 are essential amino acids stabilizing these potent compounds as found in the other SARS-CoV-2 3CL^pro^ inhibitors [40]. The prenyl group at C–2 also interacted with the L141 and N142 in the oxyanion hole, E166, and V171 (Figure 4B). 

## 3. Materials and Methods

### 3.1. General Experimental Procedures

Chemicals and general procedures: Reactions were performed in oven-dried glassware and magnetically stirred under an inert atmosphere at room temperature, unless otherwise described. Commercial α-mangostin purchased from Tokyo Chemical Industry (TCI), Tokyo, Japan, was used as a standard compound during extraction and purification and was stored at 0 °C until use. Commercial reagents were obtained from Sigma-Aldrich (St. Louis, MO, USA) and TCI. Anhydrous solvents were dried over 4 Å molecular sieves. All reactions were monitored using thin-layer chromatography (TLC) using aluminum silica gel 60 F254 (Merck). Flash column chromatography was conducted using silica gel as a stationary phase. All solvents such as methanol, ethyl acetate, dichloromethane, and hexane were distilled before use. ^1^H and ^13^C NMR spectra were obtained using a Bruker ADVANCE NEO 300-MHz NMR spectrometer. ^1^H-NMR chemical shifts (δ) and coupling constants (*J*) were given in ppm and Hz, respectively. Deuterated acetone ((CD_3_)_2_CO) served as the internal standard for both ^1^H (2.05 ppm) and ^13^C (29.92 and 206.68 ppm) NMR spectra. Accurate mass spectra were recorded on a microTOF Bruker Daltonics mass spectrometer. Melting point (MP) was measured by using Büchi B-545 melting point apparatus. 

Cultured cells, media, and reagents: 3D7 *P. falciparum* strains (wild-type drug-sensitive strain) were obtained from the Malaria Research and Reference Reagent Resource Center (MR4), BEI Resources, USA. Meanwhile, *T. brucei rhodesiense* was a gift from the London School of Hygiene and Tropical Medicine, UK. The lung cancer cell lines (H292 and H460), brain cancer cell lines (SW-1088, Lot. 70036445 and U87-MG, Lot. 70029548), and breast cancer cell lines (MDA-MB-231, Lot. 70029549 and MCF-7, Lot. 70033778) were obtained from the American Type Culture Collection (ATCC), Manassas, VA, USA. Hepatocellular carcinoma cell lines (HCT-116, Lot. 70040763) were purchased from ATCC, Manassas, VA, USA, whereas HuH-7 was purchased from the JCRB Cell Bank (JCRB, Lot. 02092022).

Roswell Park Memorial Institute (RPMI) medium 1640, Dulbecco’s modified Eagle’s medium (DMEM) high-glucose and DMEM low-glucose culture media, fetal bovine serum (FBS) and l-glutamine, penicillin/streptomycin solution, and Albumax I were procured from Gibco, Gaithersburg, MA, USA. 4-(2-Hydroxyethyl)-1-piperazineethane sulfonic acid (HEPES), bathocuproinedisulphonic acid disodium salt, l-cysteine, hypoxanthine, thymidine, sodium pyruvate, trichloroacetic acid, 2-mercaptoethanol, sulforhodamine B, and resazurin were bought from Sigma-Aldrich. Minimum Essential Media (MEM) and Earle’s balanced salt solution (EBSS) were received from Hyclone Laboratories Inc., South Logan, UT, USA. The MEM/EBSS medium and FBS for Vero cell cytotoxicity were from GE Healthcare Life Sciences, Pasching, Austria. Dimethyl sulfoxide (DMSO) was purchased from Merck Millipore, Billerica, MA, USA, or otherwise Sigma-Aldrich.

### 3.2. Extraction, Isolation, and Purification of α-Mangostin from Mangosteen Pericarp

Dried *G. mangostana* pericarp powder (500 g) was packed with the white filter cloth and macerated sequentially with hexane, ethyl acetate, methanol, and a mixture of 1:1 *v*/*v* water:methanol solution. The maceration with each solvent was performed at room temperature for 72 h. Then, the filtrate was separated and evaporated under reduced pressure. Thereafter, the process was repeated in two cycles to obtain hexane, ethyl acetate, methanol, and aqueous methanolic crude extracts. The crude extracts were monitored by TLC using standard α-mangostin as a reference. The crude extract that showed a substantial amount of α-mangostin was selected for further purification through flash column chromatography using silica as a stationary phase and a mixture of hexane and ethyl acetate solution (0%–100% *v*/*v*) as an eluent. Finally, the obtained α-mangostin (**1**) was recrystallized in ethyl acetate and characterized using spectroscopic techniques. Overall, 15.8 g of **1** was obtained as a yellow powder with a 3.2% isolation yield based on the dry weight of the mangosteen pericarp. α-Mangostin, yellow amorphous solid; ^1^H-NMR (300 MHz, Acetone-d_6_): δ ppm 13.78 (1H, s, OH), 6.81 (1H, s), 6.38 (1H, s), 5.27 (2H, m), 4.13 (2H, d, *J* = 6.4 Hz), 3.79 (3H, s), 3.35, (2H, d, *J* = 7.2 Hz), 1.80 (3H, s), 1.78 (3H, s), 1.65 (3H, s), 1.65 (3H, s); ^13^C-NMR (75 MHz, Acetone-d_6_): δ ppm 182.8, 162.9, 161.7, 157.3, 156.2, 155.7, 144.4, 138.1, 131.4, 131.4, 124.7, 123.4, 112.0, 111.0, 103.6, 102.7, 93.1, 61.3, 26.8, 25.9, 25.9, 21.9, 18.3, 17.9.

### 3.3. One-Pot Synthesis of N-Containing α-Mangostin Analogs via Smiles Rearrangement

In this section, the reaction conditions were investigated. Various experimental conditions (e.g., temperature, reaction time, alkaline base, and the equivalent of KI) were optimized. The temperature was controlled using a hotplate and microwave apparatus. Meanwhile, the effects of bases such as KOH, K_2_CO_3_, and Cs_2_CO_3_ were examined. Moreover, the amount of KI as a catalyst at 0.2 and 0.5 equivalent and the stoichiometric amount at 1 equivalent were used in this study (Table 1). An efficient condition was analyzed and chosen on the basis of the isolated yield percentage.

The general procedure was observed by dissolving **1** (0.49 mmol, 200 mg), 2-chloroacetamide (0.60 mmol, 55 mg), the base (1.23 mmol), and KI (0.01–1.00 mmol) in DMF (0.05 mM, 10 mL). The Smiles rearrangement was performed in an oven-dried round-bottomed flask. The reaction mixture was heated at 90 °C for 1 h and subsequently refluxed for another 4 h or subjected to microwave for 10 min. After the reaction showed completion through TLC monitoring, the reaction was slowly cooled to ambient room temperature; distilled water (30 mL) was added, and ethyl acetate (50 mL, three times) was used for extraction. Next, the organic layers were combined and dried over anhydrous magnesium sulfate (MgSO_4_) and then filtered and concentrated using a rotary evaporator to obtain a crude product. The crude product was purified through flash column chromatography using silica gel as a stationary phase and a solution of hexane:ethyl acetate as the eluent. Each fraction was checked by TLC using a solution of hexane:ethyl acetate (3:7 *v*/*v*) as a mobile phase. The *N*-containing α-mangostin analogs were obtained as a solid. The chemical structures of the resulting analogs **2**–**4** were characterized using spectroscopic techniques.

2-((6,8-Dihydroxy-2-methoxy-1,7-bis(3-methylbut-2-en-1-yl)-9-oxo-9H-xanthen-3-yl) oxy) acetamide (**2**); brown amorphous solid; MP 162.2–163.2 °C (from acetone); R_f_ 0.73 (hexane:ethyl acetate at 3:7 *v*/*v*); ^1^H-NMR (300 MHz, Acetone-d_6_): δ ppm 13.62 (1H, s, OH), 8.44 (1H, s), 6.44 (1H, s), 5.27 (1H, m), 5.27 (1H, m), 4.23 (2H, s), 4.13 (2H, d, *J* = 7.2 Hz), 3.84 (3H, s), 3.35, (2H, d, *J* = 7.2 Hz), 1.82 (3H, s), 1.78 (3H, s), 1.64 (3H, s), 1.64 (3H, s); ^13^C-NMR (75 MHz, Acetone-d_6_): δ ppm 182.7, 171.8, 163.3, 161.6, 155.8, 155.7, 144.4, 143.1, 137.1, 131.8, 131.4, 124.5, 123.4, 114.4, 111.2, 105.8, 103.8, 93.3, 63.1, 62.0, 26.9, 25.9, 25.9, 22.0, 18.3, 17.9; HRMS (ESI) *m*/*z* 490.1843 ([M + Na]^+^, calculated for C_26_H_29_NNaO_7_, 490.1836) and 468.2016 ([M + H]^+^, calculated for C_26_H_30_NO_7_, 468.2017).

*N*-(6,8-Dihydroxy-2-methoxy-1,7-bis(3-methylbut-2-en-1-yl)-9-oxo-9H-xanthen-3-yl)-2-hydroxy acetamide (**3**); yellow amorphous solid; MP 174.1–176.6 °C (from acetone); R_f_ 0.60 (hexane:ethyl acetate at 3:7 *v*/*v*); ^1^H-NMR (300 MHz, Acetone-d_6_): δ ppm 13.71 (1H, s, OH), 6.85 (1H, NH), 6.85 (1H, s), 6.48 (1H, s), 5.25 (1H, m), 5.25 (1H, m), 4.65 (2H, s), 4.11 (2H, d, *J* = 6.3 Hz), 3.79 (3H, s), 3.40 (2H, d, *J* = 6.6 Hz), 1.82 (3H, s), 1.79 (3H, s), 1.65 (3H, s), 1.65 (3H, s); ^13^C-NMR (75 MHz, Acetone-d_6_): δ ppm 182.9, 170.0, 162.5, 160.8, 157.7, 156.3, 156.0, 144.7, 138.1, 131.9, 131.5, 124.6, 123.6, 112.1, 112.0, 104.6, 102.7, 90.8, 68.3, 61.3, 26.9, 25.9, 25.8, 22.0, 18.2, 17.9; HRMS (ESI) *m*/*z* 490.1830 ([M + Na]^+^, calculated for C_26_H_29_NNaO_7_, 490.1836).

3,6-Diamino-1-hydroxy-7-methoxy-2,8-bis(3-methylbut-2-en-1-yl)-9H-xanthen-9-one (**4**); brown amorphous solid; MP 152.3–154.6 °C (from acetone); R_f_ 0.89 (hexane:ethyl acetate at 3:7 *v*/*v*); ^1^H-NMR (300 MHz, Acetone-d_6_): δ ppm 14.22 (1H, s, OH), 6.55 (1H, s), 6.12 (1H, s), 5.69 (2H, s, NH_2_), 5.37 (2H, s, NH_2_), 5.29 (1H, t, *J* = 6.6 Hz), 5.14 (1H, t, *J* = 6.9 Hz), 4.07 (2H, d, *J* = 6.3 Hz), 3.71 (3H, s), 3.29 (2H, d, *J* = 6.6 Hz), 1.80 (3H, s), 1.79 (3H, s), 1.66 (3H, s), 1.63 (3H, s); ^13^C-NMR (75 MHz, Acetone-d_6_): δ ppm 181.7, 161.2, 156.8, 156.0, 154.3, 148.7, 142.7, 136.8, 132.5, 130.7, 125.5, 123.0, 108.9, 106.5, 101.3, 99.1, 91.1, 60.2, 26.8, 25.9, 25.8, 22.2, 18.3, 17.9; HRMS (ESI) *m*/*z* 431.1937 ([M + Na]^+^, calculated for C_24_H_28_N_2_NaO_4_, 431.1947) and 409.2125 ([M + H]^+^, calculated for C_24_H_29_N_2_O_4_, 409.2127).

### 3.4. Cytotoxicity Evaluation against Cancer Cell Lines

The cytotoxicity was evaluated using in vitro 3-(4,5-dimethylthiazol-2-yl)2,5-diphenyltetrazolium bromide (MTT) colorimetric assay, which measures the capacity of mitochondrial enzymes (e.g., succinate dehydrogenase) present in viable cells to reduce the tetrazolium compound of MTT to its cell-membrane-impermeable purple formazan crystals. H292 and H460 were seeded in a 96-well flat-bottom microtiter plate at a cell density of 5 × 10^3^ cells/well in RPMI medium. MCF-7 and HepG2 were seeded in a 96-well flat-bottom microtiter plate at a cell density of 2 × 10^4^ cells/well in RPMI medium. SW-1088, U87-MG, MDA-MB-231, HT-29, and HCT-116 were seeded in a 96-well flat-bottom microtiter plate at a cell density of 1 × 10^4^ cells/well in a DMEM high-glucose medium. HuH7 were seeded in a 96-well flat-bottom microtiter plate at a cell density of 2 × 10^4^ cells/well in a DMEM low-glucose medium. These cell lines were allowed to adhere to the plates for 24 h at 37 °C in a 5% CO_2_ incubator. After 24 h, the cells were treated with a series of α-mangostin analogs at different concentrations and avoided the precipitation of the compounds in the cell culture media. The test compounds were made through the serial dilution concentration in the culture medium containing <1.0% DMSO. Cisplatin was used as a positive control, and the medium was applied as a negative control. The treated cells were incubated at 37 °C in a 5% CO_2_ atmosphere for 24 h. Thereafter, the cell viability was determined by adding 100 µL of the MTT solution in the culture medium at a 0.4-mg/mL concentration, and the test plates were incubated in a dark place at 37 °C in a CO_2_ incubator for 3 h. After incubation, the liquid media were removed, and 100 µL of DMSO was added into each well to dissolve the formazan purple crystals. The intensities of the dissolved formazan crystals and the absorbance were measured using a microplate reader at 570 nm. Then, the percentage of inhibition and IC_50_ values of the cells were calculated from nonlinear regression analysis. Each experiment was performed in triplicate.

### 3.5. In Vitro Antibacterial Assay

The antibacterial activity was determined according to the broth microdilution procedure using standard 96-well microtiter plates [42]. Four Gram-positive bacteria (*S. aureus* ATCC 25923, *S. epidermidis* ATCC 12228, *K. rhizophila* ATCC 9341, and *B. subtilis* ATCC 6633) and two Gram-negative bacteria (*K. pneumoniae* ATCC 13883 and *E. coli* ATCC 25922) were used as the tested microorganisms. The tested compounds were dissolved in DMSO and subsequently prepared for serial twofold dilution (50 µL) using the Mueller–Hinton Broth (MHB) in a microtiter plate. The fresh inoculum, obtained from the preculture of tested bacteria in MHB at 37 °C for 18–24 h, was adjusted to 1 × 10^8^ cfu/mL and then diluted to 1:100 *v*/*v*. Finally, 50 µL of the bacterial suspension was added into each well previously loaded with 50 µL of the tested compounds to give the final volume of 100 µL. After inoculation, each well contained bacterial cells at approximately 5 × 10^5^ cfu/mL. The microtiter plate was then incubated at 37 °C for 24 h. The lowest dilution of each compound with no bacterial growth was recorded as the MIC. Vancomycin was used as the positive control for the tested Gram-positive bacteria, whereas erythromycin and gentamicin were used as the negative control for Gram-negative bacteria.

### 3.6. In Vitro Antimalarial Assay

Parasite *P. falciparum* strains 3D7 (a wild-type drug-sensitive strain) was used in this study. This parasite was maintained continuously in human O^+^ erythrocytes at 37 °C under 3% CO_2_ and 90% N_2_ in RPMI 1640 culture media (Life Technologies Limited, Paisley, UK) supplemented with 2-mM l-glutamine, 25-mM HEPES, pH 7.4, 0.2% NaHCO_3_, 40-mg/L gentamicin, 0.37-mM hypoxanthine, and 5-g/L Albumax I (Life Technologies, Grand Island, NY, USA). In vitro antimalarial activity was determined using the malaria SYBR green I-based fluorescence method. Briefly, 0.09 mL of cultured 1% ring-stage synchronized parasites were transferred to individual wells of a standard 96-well microtiter plate, and in vitro culture continued for 48 h, with 0.01 mL of the test compound at different concentrations in each well. The compounds were dissolved in DMSO, and the final concentration of DMSO in each well was 0.1%, which had no effect on parasite viability. After 48 h, SYBR Green I was added to each well, and fluorescence signals were measured using a spectrofluorometer at wavelength ex485/em535 nm. The results were recorded as the concentration of each compound that exhibited 50% growth inhibition (IC_50_) from the dose–response curve established from the fluorescence signals at each compound concentration [43].

### 3.7. In Vitro Antitrypanosomal Assay

*Trypanosoma brucei rhodesiense* (STIB-900) was maintained continuously in a MEM/EBSS medium (Hyclone Laboratories Inc., South Logan, Utah) supplemented with 3.0-g/L NaHCO_3_, 4.5-g/L glucose, 25-mM HEPES, pH 7.3, 0.05-mM bathocuproinedisulphonic acid disodium salt, 1.5-mM l-cysteine, 1-mM hypoxanthine, 0.16-mM thymidine, 1-mM sodium pyruvate, 0.2-mM 2-mercaptoethanol, 1% MEM non-essential amino acid, and 15% *v*/*v* heated FBS in a 5% CO_2_ incubator [44]. Parasites at the density of 2 × 10^4^
*Trypanosoma* cells per well were incubated with serial concentrations of each compound in a 96-well plate under the same culture conditions to evaluate the antitrypanosomal activity. The compounds were dissolved in DMSO, and the final concentration of DMSO in each well was 0.1%, which had no effect on parasite viability. After 72 h of incubation, 20 µL of resazurin was added to each well. The reaction was further incubated at 37 °C and 5% CO_2_ for 3 h to allow the irreversible reduction of resazurin (violet color) to resorufin (pink color) by viable *Trypanosoma* cells. The fluorescence signals were measured using a spectrofluorometer at wavelength ex 530/em 585 nm. The results were recorded as the concentration of each compound that exhibited 50% growth inhibition (IC_50_) from the dose–response curve established from the fluorescence signals at each compound concentration.

### 3.8. In Vitro SARS-CoV-2 Main Protease (3CL^pro^) Inhibition Assay

The SARS-CoV-2 3CL^pro^ inhibition assay was performed following a reported procedure [37]. The recombinant 3CL^pro^ was prepared using *E. coli*, as previously reported [45]. The assay was performed with 0.2-µM 3CL^pro^ in the presence of 25-µM E(EDANS)TSAVLQSGFRK(DABCYL) as the fluorogenic substrate in a phosphate buffer solution consisting of 1.0-mM dithiothreitol and 2% DMSO. The volume of each reaction was fixed at 100 µL. The fluorescence signals were measured using a microplate reader (H1; BioTek Synergy) at ex340/em490-nm wavelengths. The initial rates (relative fluorescence unit per second [RFU/s]) were measured in the absence and presence of compounds at 100 µM. The inhibition results were displayed as relative percentages compared with the initial rate in the absence of the inhibitor. Next, the test compounds were further evaluated for the effective half-maximal inhibitory concentrations (IC_50_) using serial concentrations at 10–500 μM. Each test was performed in triplicate. Data were shown as mean and standard error (S.E.) of three biological independent experiments.

### 3.9. In Silico Study of Potent Compounds toward SARS-CoV-2 Main Protease (3CL^pro^)

The crystal structure of SARS-CoV-2 3CL^pro^ in complex with a noncovalent inhibitor (X77) derived from the protein databank (PDB code: 6W63) [46] was used to generate the complex structure with a potent compound by using the molecular docking [47,48] and subsequent FMO calculation according to the standard protocols used in our previous studies [39,49]. The 3D structures of potent compounds; α-mangostin (**1**), analogs **2**, and **4** were constructed using GaussView6 software and structurally optimized by the B3LYP/6-31G* level of theory by the Gaussian16 program [50]. The electrostatic potential charges (ESP) of the optimized compounds obtained from the same level of theory were converted to the restrained electrostatic potential (RESP) charges by the antechamber in AMBERTools 21 [51]. All missing hydrogen atoms of the docked complex were added using tLeap and then minimized by steepest descent and conjugate gradient methods using the sander module implemented in the AMBER20 package program [51]. The resulting structure of ligand/3CL^pro^ complex was retrieved for calculating the pair interaction energy (PIE) and decomposition (PIEDA) using FMO and the resolution of the identity second-order Møller–Plesset perturbation theory (RI-MP2) combined with PCM solvation, as described in our previous works [37,52]. The interaction profile of the potential compounds in complex with 3CL^pro^ was represented by a stacked bar graph and grid map based on PIE and PIEDA data. The UCSF Chimera V1.15 [53]. BIOVIA Discovery Studio Visualizer [41] were used to visualize all 3D and 2D structures, respectively.

## 4. Conclusions

The one-pot synthesis of new *N*-containing α-mangostin analogs via Smiles rearrangement was accomplished using Cs_2_CO_3_ in the presence of 2-chloroacetamide and catalytic KI to obtain ether **2**, amide **3**, and amine **4** in good yield (85%) at an optimum ratio of 1:3:3. The intramolecular nucleophilic aromatic substitution of α-mangostin occurred regioselectively at the hydroxyl groups at C–3 and C–6. Compound **3** contained a 2-hydroxy acetamide substituent at C–3, whereas compound **4** had amine groups at C–3 and C–6. The new *N*-containing α-mangostin analogs **3** and **4** displayed the isosteric replacement of the oxygenated motif with nitrogenated substituents, enhancing the compound’s flexibility, H-bond acceptors, H-bond donors, and solubility. Moreover, the biological activities of xanthones **3** and **4** were reported for the first time. Amine **4** exhibited cytotoxicity against the highly metastatic H460 non-small-cell lung cancer cell line at IC_50_ 13.89 ± 2.55 µM, which was more potent than those of **1** and cisplatin by twofold and fourfold, respectively. Xanthones **2**–**4** showed mild antibacterial activity against the tested Gram-positive bacteria, especially *K. rhizophila*, with IC_50_ values of 9.76, 78.13, and 4.88 µM, respectively. The 3,6 diamine-α-mangostin analog **4** revealed an interesting antibacterial activity with bacterial membrane permeation compared with those of ether **2** and amide **3**. Among the xanthones in this series, amide **3** exhibited strong antimalarial activity against *P. falciparum* at IC_50_ 4.25 ± 0.60 µM, whereas both amide **3** and amine **4** displayed antitrypanosomal activity against *Trypanosoma brucei rhodesiense* at IC_50_ values of 2.68 ± 0.57 and 2.32 ± 0.28 µM, respectively, which were twofold more potent than that of **1**, a mother compound. Interestingly, compound **2** exhibited potent SARS-CoV-2 main protease 3CL^pro^ inhibition at 24.6 ± 1.1 µM, which better that **1** and rutin approximately three- and fourfold, whereas compound **4** showed similar inhibitory activity to **1**. The fragment molecular orbital (FMO) method indicated the unique binding interaction of the xanthone core at 3CL^pro^ active site. The improved binding interaction of **2** involved the interaction of an additional ether moiety containing the amide group with M49, L50, Y54, and R188 amino acid residues in the 3CL^pro^ active site. The results suggest that *N*-containing α-mangostin analogs derived from the isosteric replacement of the hydroxyl groups at C–3 and C–6 via Smiles rearrangement possessed advanced biological activity, mediated by the improved druglike properties. Therefore, *N*-containing xanthone analogs would be further studied as potential biotically active agents for anti-lung-cancer, antitrypanosomal, and anti-SARS-CoV-2 main protease (3CL^pro^) inhibitory activities.

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
