# Peer review of "N-Containing α-Mangostin Analogs via Smiles Rearrangement as the Promising Cytotoxic, Antitrypanosomal, and SARS-CoV-2 Main Protease Inhibitory Agents"

_molecules, 2023, doi:10.3390/molecules28031104_

Round 1

Reviewer 1 Report

Authors present an interesting Smiles rearrangements and the synthesis and biological evaluation of several compounds. The chemistry part is solid, the biological results are somewhat questioned. 

1. It is suggested to cite the work of Mayer et al. describing a very similar reaction: https://www.mdpi.com/1420-3049/25/4/888

2. In Table 2 it is not clear, why isthere ">50", ">100 " written, if in other cases more exact values around this value is shown e.g. 72.55

3. It seems that the new analogues generally lost citotoxic activity compared to Mangostin. And unfortunately none of the compounds are selective, the Vero cells are also destroyed with very similar activity. It is suggested to remove the citotoxicity results, or interpret very carefully.

4. The antibacterial results should contain the >2.5 mM and not "-". Unfortunately th resuls are again not convicing, the new derivatives are less active. 

5. Table 4 shows convincing results, the structural change did not decrease the potency generally. 

6. Authors are kindly asked to explain why rutin, and not a more potent compound was chosen as control against Sars-Cov2. It is suggested to show IC50 values.

Author Response

Regarding reviewer 1 comments, we sincerely appreciate your suggestions to improve the scientific quality of the manuscript. Therefore, we would like to address your comments in the following responses. The revisions were shown in the manuscript with a yellow highlight.

  1. It is suggested to cite the work of Mayer et al. describing a very similar reaction: https://www.mdpi.com/1420-3049/25/4/888

Response: We added a sentence mentioning the transformation of chrysin, a natural flavonoid, to 7-aminochrysin derivatives by Smiles rearrangement in lines 110-112 along with the additional suggested reference by Mayer as shown in the reference 29 in lines 988-989.

  1. In Table 2 it is not clear, why is there ">50", ">100 " written, if in other cases more exact values around this value is shown e.g. 72.55

Response: The Cytotoxicity was further calculated to obtain specific IC50 in the micromolar range, as shown in Table 2 at lines 265-268.

  1. It seems that the new analogs generally lost cytotoxic activity compared to Mangostin. And unfortunately, none of the compounds are selective, the Vero cells are also destroyed with very similar activity. It is suggested to remove the cytotoxicity results, or interpret very carefully.

Response: The cytotoxicity of compounds 1-4 was elaborated in lines 256-264. a-Mangostin (1) showed the most interesting cytotoxicity among its analogs in this series. However, compound 3, a new a-mangostin analog, exhibited cytotoxic activity stronger than cisplatin against brain, breast, hepatocellular, and colon cancer cell lines (lines 256-258). Notably, compound 4, which is also a new a-mangostin analog, showed promising cytotoxicity against the non-small-cell lung cancer H460 cell line with twofold and fourfold more potent than that of a-mangostin and cisplatin, respectively (lines 258-264). Since compounds 3 and 4 are new N-containing a-mangostin analogs, their biological activities have never been reported. Therefore, we would like to show their cytotoxic profile against several cancer cell lines.

Regarding the cytotoxicity against Vero cells, we agreed with the reviewer that compounds 1-4 showed similar cytotoxicity. Thus, we removed cytotoxicity results against Vero cells, as the reviewer suggested.

Further studies on drug-like properties and anticancer mechanisms are required to verify the benefit of the new compounds 3 and 4. This remark was added in lines 262-264.

  1. The antibacterial results should contain the >2.5 mM and not "-". Unfortunately, the results are again not convincing, the new derivatives are less active.

Response: The result of Table 3 was revised based on the reviewer’s comment. MIC value  >2.5 mM was shown in Table 3 in lines 349-350.

  1. Table 4 shows convincing results, the structural change did not decrease the potency generally.

Response: We agreed. The series of N-containing a-mangostin analogs 3 and 4 exhibited improved antitrypanosomal activity against Trypanosoma brucei rhodesiense at threefold stronger potency than that of a-mangostin.

  1. Authors are kindly asked to explain why rutin, not a more potent compound, was chosen as a control against Sars-Cov2. It is suggested to show IC50 values.

Response: Retin was reported recently by our team that it is an inhibitor of the 3CLPro [Ref. 37]. Using rutin is convenient because it is commercially available, while the other potent competitive inhibitors reported in the literature are unavailable. Other potent inhibitors that are commercially available, such as nirmatrelvir, covalently inhibit 3CLPro. Therefore, they exhibit time-dependent inhibition and may not be suitable as a positive control in our experiment [Ref. 37] for initial screening or comparison with other inhibitors that might be reversible and competitive.

As the reviewer’s suggested, we performed the additional in vitro SARS-CoV-2 main protease (3CLPro) inhibition assay to obtain IC50 values, as shown in Figure 3 and Table 5 at lines 464-486. The inhibitory activity data was described in lines 456-463.

Ref. 37:

Deetanya, P., Hengphasatporn, K., Wilasluck, P., Shigeta, Y., Rungrotmongkol, T., & Wangkanont, K. (2021). Interaction of 8-anilinonaphthalene-1-sulfonate with SARS-CoV-2 main protease and its application as a fluorescent probe for inhibitor identification. Computational and Structural Biotechnology Journal, 19, 3364-3371.

Reviewer 2 Report

The authors described alfa-Mangostin analogs synthesis and biological potency evaluation. My feeling is, the synthetic methodology is not new and there are no significant results of all activity tests with the author's 3 compounds. Indeed, it is difficult to recommend this manuscript to publish in any journals at this stage. I recommend that the authors should comment either e expected targets or computational study results for being intersting for readers.

Author Response

Regarding the reviewer’s comment toward the addition of computational study, we respectively appreciated and agreed. Overall, the most interesting biological activity of N-containing a-mangostin analogs is the inhibition of SARS-CoV-2 targeted 3C-like main protease (3CLPro). We performed the additional in vitro 3CLPro inhibitory assay to obtain IC50 values, as shown in Figure 3 and Table 5 at lines 464-486. The inhibitory activity data was described in lines 456-463. Compound 2 showed potent inhibition at IC50 24.6 ± 1.1 mM, threefold better than mother compound 1. Furthermore, compounds 1 and 4 exhibited similar inhibitory activity at IC50 61.6 ± 1.1 and 63.8 ± 1.2 mM, respectively.

Therefore, we perform a computational study including docking and fragment molecular orbital (FMO) method toward the 3CLPro to predict the binding mode of a-mangostin (1), ether analog 2, and diamine analog 4, as shown in Figure 4.  The binding energy, binding profile,  pair interaction energy (PIE), and its decomposed analysis (PIEDA) at the FMO-RIMP2/PCM level of the potent compounds toward SARS-CoV-2 3CLPro was explained in lines 488-509. The detailed in silico experiment was described in lines 806-840.

We observed that the xanthone core structure of these compounds is likely aligned into the same binding pocket (S1, S2, and S1') at the 3CLpro active site. This core structure at the methoxy group at carbon position 7 and the prenyl group at carbon position 8 mainly interacted with the residues 41-45 in the S2 subpocket. Interestingly, the N-containing compound 2 showed an interaction energy of -106.23 kcal/mol, significantly lower than its parent compound 1 (-72.09 kcal/mol) and analog 4 (-69.35 kcal/mol). The additional substituent containing an ended primary amide group at carbon position 6 of compound 2, obtained from Smile rearrangement, enhances binding interaction by interacting with M49, L50, Y54, and R188 with three H-bond formations at M49, Y54, and D187. Moreover, the catalytic residues H41 and E166 are essential amino acids stabilizing these potent compounds as found in the other SARS-CoV-2 3CLPro inhibitors.

The revisions were shown in the manuscript with a yellow highlight.

Reviewer 3 Report

This manuscript by Chamni group reported a one-pot synthesis of new N-containing -mangostin derivatives such as its ether, amide, and amine via Smiles rearrangement with the aid of Cs2CO3 in the presence of 2-chloroacetamide and catalytic KI. Also, the authors evaluated their biological activities such as SARS-CoV-2 protease inhibitory activity, antimalarial, antibacterial, and anticancer activities. The described studies could be an interesting extension of the current knowledge and well-established by others. overall, I have only a few questions that need to be justified before its acceptance.

Comments

1.     Title needs to be changed with a better view.

2.     The authors should describe previous reports with this including citations as a figure.

3.     Figure 1 changed as Scheme 1 and the formation of those derivatives is explained in detail (mechanisms).

4.     NMR analysis of these derivatives should be included with structures as a figure for better understanding as per the reader’s view in the main article instead of the SI file.

5.     MPs of these scaffolds must be provided in the exp. section.

6.     In SI file, NMR integrations of compound 2 and 3 needs attention (proton count)

7.     I can see that several impurities need to be addressed whether it is solvent or something else. Purity is the main imp—task as per biology perspective.

8.     Authors provided 2D NMR and please include with analysis in the provided copies.

9.     NMR data is not consistent with provided data in the main article (1H/13C) and must be checked thoroughly.

10.  Several places in the article contain errors and typos and to be rechecked again.

Author Response

Regarding reviewer 3 comments, we sincerely appreciate your suggestions to improve the scientific quality of the manuscript. Therefore, we would like to address your comments in the following responses. The revisions were shown in the manuscript with a yellow highlight.

  1. Title needs to be changed with a better view.

Response: We appreciate your comment and agree to revise the title of the manuscript to “N-Containing a-Mangostin Analogs via Smiles Rearrangement as the Promising Cytotoxic, Antitrypanosomal, and SARS-CoV-2 Main Protease Inhibitory Agents” as shown in lines 1-4

  1. The authors should describe previous reports with this including citations as a figure.

Response: We believe you mean examples of N-containing a-mangostin analogs at C–3 and C–6 positions obtained from the Mannich reaction and carbamate formation, as mentioned in line 99. Therefore, we added the new figure as Figure 1 to showed reported N-containing a-mangostin analogs corresponded with references 17, 18, and 22-25 (lines 115-117).

  1. Figure 1 changed as Scheme 1 and the formation of those derivatives is explained in detail (mechanisms).

Response: We changed “Figure 1” to “Scheme 1”, as shown in line 168. The proposed mechanism for the Smiles rearrangement of a-mangostin was briefly explained in lines 137-144 and illustrated in Scheme 2 in lines 180-184.

  1. NMR analysis of these derivatives should be included with structures as a figure for better understanding as per the reader’s view in the main article instead of the SI file.

Response: 1H and 13C NMR chemical shift in acetone-d6 (given in ppm) of compound 1-4 were added as Figure 2 in lines 185-188.

  1. MPs of these scaffolds must be provided in the exp. section.

Response: General protocol for melting points (MPs) measurement was added in line 535. We performed an experiment to evaluate the melting point of compounds 2-4. The melting point of compound 2 is shown in line 606. The melting point of compound 3 is shown in line 630. The melting point of compound 4 is shown in line 639.

  1. In SI file, NMR integrations of compound 2 and 3 needs attention (proton count)

Response: H-NMR spectra of compounds 2 and 3 were updated and showed data of integrations and chemical shifts as shown in Figures S3 and S8.

  1. I can see that several impurities need to be addressed whether it is solvent or something else. Purity is the main imp—task as per biology perspective.

Response: Deuterated acetone ((CD3)2CO) was used as an NMR solvent and served as the internal standard for both 1H (2.05 ppm) and 13C (29.92 and 206.68 ppm). The peak at 2.81 ppm is the peak of HDO that is commonly observed when using deuterated acetone. Impurities found in NMR spectra of compounds 2 and 3 correspond to the minor products of Smile rearrangement that occurred at the C-3 position.

  1. Authors provided 2D NMR and please include with analysis in the provided copies.

Response: We revised 2D NMR spectra by adding the analysis shown in Supporting Information.

  1. NMR data is not consistent with provided data in the main article (1H/13C) and must be checked thoroughly.

Response: We appreciate your comment and checked the manuscript and Supporting Information thoroughly. The errors and typos were edited in lines 578, 580, 632, 633, 635, and 640 with yellow highlights.

  1. Several places in the article contain errors and typos and to be rechecked again.

Response: We appreciate your comment and checked the manuscript thoroughly. The errors and typos were edited as follows.

Line 43  Trypanosome changed to Trypanosoma

Line 83  Trypanosome changed to Trypanosoma

Line 278 Add mM

Line 360 Table 4 Antimalaria changed to Antimalarial and Antitrypanosoma changed to Antitrypanosomal

Line 447 activities changed to activity

Line 447 inhibition changed inhibitory

Line 449 antitrypanosomals and antitrypanosomal agents

Line 449 espectially as changed to notably

Line 449 Trypanosome changed to Trypanosoma

Line 450 plasmodium changed to Plasmodium

Line 708  Trypanosome changed to Trypanosoma

Line 851  Trypanosome changed to Trypanosoma

Round 2

Reviewer 1 Report

Authors answered all the questions and comments and the manuscript was developed significantly. Re-thinking a bit and with the aim of being open and honest I suggest to include a sentence sharing the citotoxicity results against Vero cells (and move the data to the supplementary).  This is an important finding for those would use this paper as a basis of research, and sharing this result emphasizes the option for further development of these compounds.

Author Response

Thank you for your kind support. We appreciate your thoughts and consideration. 

We agreed to add cytotoxicity evaluation against the Vero cell line. We added an explanation in the manuscript with a blue highlight as follows:

Lines 235-237: add a sentence “Furthermore, compounds 1 and 3 showed cytotoxic activities against Vero cells at EC50 22.00 ± 1.44 µM and 26.48 ± 0.59 µM respectively, whereas compounds 2 and 4 had low cytotoxicity (see Supporting Information, Table S4)”.

Line 611-612: add text “cytotoxicity evaluation against the Vero cell line and experimental procedure (see Table S4)”.

Moreover, the result and experimental detail were also added in the Supporting Information, as shown on page S16 and Table S4.

Reviewer 2 Report

The authors have already corrected and answered all questions. Indeed, I agree that this manuscript can be published in Molecules.

Author Response

Thank you for your kind support. We appreciate your thoughts and consideration.

Reviewer 3 Report

The authors have improved the quality of the manuscript and solved my previous concerns and this revised version can be acceptable for publication.

Author Response

(The authors gave the same response as above.)
